# Mammalian Cell-Free System Recapitulates the Early Events of Post-Fertilization Sperm Mitophagy

**DOI:** 10.3390/cells10092450

**Published:** 2021-09-17

**Authors:** Won-Hee Song, Dalen Zuidema, Young-Joo Yi, Michal Zigo, Zhibing Zhang, Miriam Sutovsky, Peter Sutovsky

**Affiliations:** 1Division of Animal Science, University of Missouri, Columbia, MO 65211, USA; whsong85@gmail.com (W.-H.S.); dmzfdb@mail.missouri.edu (D.Z.); yiyj@scnu.ac.kr (Y.-J.Y.); zigom@missouri.edu (M.Z.); SutovskyM@missouri.edu (M.S.); 2Department of Agricultural Education, College of Education, Sunchon National University, Suncheon 57922, Korea; 3Department of Physiology, Wayne State University, Detroit, MI 48201, USA; gn6075@wayne.edu; 4The C.S. Mott Center for Human Growth and Development, Department of Obstetrics & Gynecology, Wayne State University, Detroit, MI 48201, USA; 5Department of Obstetrics, Gynecology and Women’s Health, University of Missouri, Columbia, MO 65211, USA

**Keywords:** mitochondria, mtDNA, cell-free system, ubiquitin-proteasome system, autophagy, SQSTM1, VCP, PACRG, SPATA18

## Abstract

Propagation of paternal sperm-contributed mitochondrial genes, resulting in heteroplasmy, is seldom observed in mammals due to post-fertilization degradation of sperm mitochondria, referred to as sperm mitophagy. Whole organelle sperm mitochondrion degradation is thought to be mediated by the interplay between the ubiquitin-proteasome system (UPS) and the autophagic pathway (Song et al., Proc. Natl. Acad. Sci. USA, 2016). Both porcine and primate post-fertilization sperm mitophagy rely on the ubiquitin-binding autophagy receptor, sequestosome 1 (SQSTM1), and the proteasome-interacting ubiquitinated protein dislocase, valosin-containing protein (VCP). Consequently, we anticipated that sperm mitophagy could be reconstituted in a cell-free system consisting of permeabilized mammalian spermatozoa co-incubated with porcine oocyte extracts. We found that SQSTM1 was detected in the midpiece/mitochondrial sheath of the sperm tail after, but not before, co-incubation with oocyte extracts. VCP was prominent in the sperm mitochondrial sheath both before and after the extract co-incubation and was also detected in the acrosome and postacrosomal sheath and the subacrosomal layer of the spermatozoa co-incubated with extraction buffer as control. Such patterns are consistent with our previous observation of SQSTM1 and VCP associating with sperm mitochondria inside the porcine zygote. In addition, it was observed that sperm head expansion mimicked the early stages of paternal pronucleus development in a zygote during prolonged sperm-oocyte extract co-incubation. Treatment with anti-SQSTM1 antibody during extract co-incubation prevented ooplasmic SQSTM1 binding to sperm mitochondria. Even in an interspecific cellular environment encompassing bull spermatozoa and porcine oocyte extract, ooplasmic SQSTM1 was recruited to heterospecific sperm mitochondria. Complementary with the binding of SQSTM1 and VCP to sperm mitochondria, two sperm-borne pro-mitophagy proteins, parkin co-regulated gene product (PACRG) and spermatogenesis associated 18 (SPATA18), underwent localization changes after extract coincubation, which were consistent with their degradation observed inside fertilized porcine oocytes. These results demonstrate that the early developmental events of post-fertilization sperm mitophagy observed in porcine zygote can be reconstituted in a cell-free system, which could become a useful tool for identifying additional molecules that regulate mitochondrial inheritance in mammals.

## 1. Introduction

Clonal, maternal inheritance of mitochondria and mitochondrial DNA is observed in humans and prevalent in most animals [1]. Sperm mitophagy, the elimination of sperm-borne mitochondria after fertilization, assures normal preimplantation development and might prevent adverse effects on fitness and fertility arising from heteroplasmy, the coexistence of two distinct mitochondrial genomes in one organism [2,3,4,5]. Early studies of mammalian sperm mitophagy reported that the sperm mitochondrial sheath becomes ubiquitinated during spermatogenesis and surrounded by the oocyte-derived lysosome-like structures after fertilization, thus implicating both the ubiquitin-proteasome system (UPS) and the autophagic pathway in the elimination of paternal mitochondria [6,7]. The porcine zygote is a particularly useful model system as sperm mitophagy is observed prior to the first embryo cleavage, earlier than in other relevant mammalian models. Cell-free systems are yet to be applied to this area of research, relying on monitoring mitochondria from one spermatozoon per zygote. While frog egg extracts are commonly used in cell biology, they are not suitable for the study of mammalian sperm mitophagy, which appears to be species-specific and not observed in interspecific crosses. Our recent observations in zygotes of the domestic pig and rhesus monkey indicate that post-fertilization sperm mitophagy depends on both autophagy driven by ubiquitin-binding autophagy receptor, sequestosome 1 (SQSTM1). Presentation of ubiquitinated sperm mitochondrial membrane proteins to the 26S proteasome is mediated by the ubiquitinated protein dislocase, valosin-containing protein (VCP) [8]. SQSTM1 was found to exclusively associate with boar and rhesus monkey sperm mitochondria after fertilization. While VCP was detectable in the sperm midpiece, both before and after in vitro fertilization, transient pharmacological inhibition of VCP function delayed the progress of sperm mitophagy in porcine zygotes. A combined treatment interfering concomitantly with SQSTM1 and VCP in the zygotes delayed sperm mitophagy up to the two–four-cell stages of porcine embryonic development, at which time the control zygotes no longer contained detectable sperm mitochondria. These findings raised the question of whether the post-fertilization sperm mitophagy could be reconstituted in a cell-free system, an in vitro research tool that could simplify the identification of additional mitophagy regulators.

Cell-free systems are widely used to study biochemical and molecular events, which reduce the complexity of interactions seen in whole cells [9]. Cell-free systems derived commonly from *Xenopus*
*laevis*, or rarely from mammalian oocytes have been used to reconstitute early events of zygotic development [9,10,11]. The large size of *Xenopus* oocytes assures large quantities of cell extract suitable for analyzing molecular and subcellular events occurring at low detection levels in a zygote fertilized by a single spermatozoon. For example, the assembly and incorporation of the nuclear pore complexes (NPC) into the nuclear envelope formed *de novo* around the demembranated bull sperm nuclei was reconstituted in *Xenopus* oocyte extracts [11]. Under cell-free conditions, treatment with microtubule inhibitors reduced the assembly of NPCs around the sperm nuclei as well as sperm head expansion that signals the onset of paternal pronuclear development. The morphological events observed under cell-free conditions thus replicated NPC assembly observed during natural fertilization. The *Xenopus* oocyte extracts have also been used to induce nuclear reprogramming of differentiated mammalian somatic cells. Expression of pluripotent marker genes was confirmed in the differentiated mammalian cells treated with *Xenopus* oocyte extracts [10]. In addition, cell-free extracts derived from mammalian oocytes were used to induce nuclear reprogramming in somatic cells [9,10]. A semi-cell-free system composed of live, capacitated boar spermatozoa and solubilized porcine zona pellucida (ZP) proteins were used to study the role of sperm-borne proteasomes in the digestion of ZP proteins, mimicking the events of sperm-ZP penetration during natural fertilization [12]. Together, such studies document the feasibility of reconstituting early developmental events in a cell-free system consisting of permeabilized mammalian spermatozoa co-incubated with intra- or inter-specific oocyte extracts.

Porcine oocytes are obtainable in large numbers and extract volumes, sufficient to treat thousands of spermatozoa in one trial. Thus, cell-free systems lend themselves to large-scale studies of post-fertilization sperm mitophagy, which is limited to one spermatozoon/sperm mitochondrial sheath per zygote in a conventional monospermic IVF system. Our cell-free system is able to mimic the proteomic interactions taking place between the fertilizing spermatozoa and the ooplasm during natural fertilization. The priming process of removing the plasma membrane and reducing disulfide bonds mimics spermatozoa processing which takes place prior to sperm-ooplasm contact. At that point, the demembranated and disulfide-reduced spermatozoa would enter the ooplasm, and proteomic interactions such as pronuclear development and sperm mitochondrial sheath degradation would begin. Our unique and novel cell-free system was designed with capturing these protein-protein interactions in mind. Here, we established a cell-free system consisting of permeabilized boar and bull spermatozoa co-incubated with cell extracts from porcine metaphase II (MII) oocytes and used it to examine the binding of oocyte-derived mitophagy factors, SQSTM1 and VCP, to sperm mitochondria. Furthermore, our cell-free system was used to study the localization of candidate sperm-borne mitophagy determinants, parkin co-regulated gene (PACRG), and spermatogenesis associated 18 (SPATA18) in spermatozoa before and after coincubation with the oocyte extract.

## 2. Materials and Methods

### 2.1. Semen Preparation

Boars were raised at the University of Missouri swine farm. Fresh boar semen was collected in one regular collection per week, transferred into 15 mL centrifuge tubes, and centrifuged at 800× *g* for 10 min to separate spermatozoa from seminal plasma. Sperm concentration was assessed by using a light microscope and a hemocytometer (ThermoFisher Scientific, Houston, TX, USA). Only semen collections with acceptable sperm motility were used. Spermatozoa were diluted with BTS extender (IMV Technologies, Maple Grove, MN, USA) to a final concentration of 1 × 10^8^ spermatozoa/mL and stored in a styrofoam box at room temperature for up to 5 days.

Cryopreserved bull semen was thawed at 37 °C for 40 s and washed by centrifugation in HEPES-buffered Tyrode lactate medium containing 0.01% (*w*/*v*) polyvinyl alcohol (TL-HEPES-PVA, pH = 7.4) [13]. The frozen-thawed bull spermatozoa were used without dilution.

### 2.2. Sperm Priming for Cell-Free System

Boar or bull spermatozoa were washed with phosphate-buffered saline (PBS, 137 mM NaCl, 2.7 mM KCl, 10 mM, 11 Na_2_HPO_4_, 1.8 mM KH_2_HPO_4_, pH = 7.2) containing 0.1% PVA (PBS-PVA) two times by centrifugation at 800× *g* for 5 min. To stain sperm mitochondria, the spermatozoa were labeled with a fixable, vital, mitochondrion selective probe MitoTracker^®^ Red CMXRos (Invitrogen-Molecular Probes, Eugene, OR, USA) for 10 min at 37 °C. At the previously tested concentration of 400 nM, the probe specifically stains bull sperm mitochondria, while in the boar, the probe is also taken up by the sperm head structures [8]. However, this is not detrimental to the colocalization of autophagy proteins with sperm mitochondria in zygotes or the cell-free system described here.

Sperm heads and tails were separated to examine sperm morphological changes after oocyte extract co-incubation. Spermatozoa pre-labeled with MitoTracker were sonicated with a digital sonifier (Branson) at 30% intensity for 1 min in RIPA buffer (200 mM Tris∙HCI, pH = 8; 150 mM NaCI; 1% (*v*/*v*) Triton X-100; 0.5% (*w*/*v*) sodium deoxycholate; 0.1% (*w*/*v*) SDS; and 1 mM PMSF) [14].

To prime sperm mitochondrial sheaths for cell-free studies, spermatozoa pre-labeled with MitoTracker were demembranated/permeabilized with 0.05% (*w*/*v*) lysolecithin (L-α-lysophosphatidylcholine, Sigma, St. Louis, MO, USA) in KMT (20 mM KCl, 5 mM MgCl_2_, 50 mM TRIS∙HCl, pH = 7.0) for 10 min at 37 °C, and washed twice with the KMT for 5 min by centrifugation, to terminate the reaction. The spermatozoa were subsequently incubated with 0.0, 0.1, 1.0, and 10.0 mM dithiothreitol (DTT; disulfide bonds/S-S reduction agent; Sigma, St. Louis, MO, USA) diluted in KMT, pH = 8.2 for 20 min at 37 °C and washed twice with KMT for 5 min by centrifugation, to terminate the reaction. Such sequential lysolecithin-DTT treatment was applied to mimic the demembranation and S-S reduction in the sperm mitochondrial sheath following natural fertilization [15].

### 2.3. In Vitro Maturation of Porcine Oocytes

Porcine ovaries were obtained from a local slaughterhouse. Cumulus-oocyte complexes (COCs) were aspirated from ovarian follicles sized 3–6 mm, divided into groups of 100–150 in 4-well culture dishes (Nunc, Roskilde, Denmark), and matured in TCM-199 medium (Mediatech, Inc., Manassas, VA, USA) supplemented with 0.1% PVA, 3.05 mM D-glucose, 0.91 mM sodium pyruvate, 0.57 mM cysteine, 0.5 μg/mL FSH, 0.5 μg/mL LH, 10 ng/mL EGF, 10% porcine follicular fluid, 75 μg/mL penicillin G and 50 μg/mL streptomycin. The COCs were incubated at 38.5 °C, with 5% CO_2_ in the air. After 22 h of culture, the COCs were transferred into TCM199 medium without FSH and LH and then cultured for an additional 22 h at 38.5 °C and 5% CO_2_ in the air.

### 2.4. In Vitro Fertilization (IVF) and In Vitro Culture (IVC) of Pig Oocytes/Zygotes

Cumulus cells from matured COCs were removed with 0.1% hyaluronidase in TL-HEPES-PVA medium and washed three times with TL-HEPES-PVA medium. The oocytes were washed one more time with Tris-buffered medium (mTBM) containing 0.3% BSA (A7888, Sigma). Between 25–30 oocytes/drop were placed into 100 µL drops of the mTBM covered with mineral oil in a 35 mm polystyrene culture dish, then incubated until spermatozoa were prepared for fertilization. Liquid semen preserved in BTS extender solution was washed with PBS containing 0.1% PVA (PBS-PVA) two times by centrifugation at 800× *g* for 5 min. To stain mitochondria in the sperm tail, the boar spermatozoa were incubated with vital, fixable, mitochondrion-specific probe MitoTracker^®^ Red CMXRos (Molecular Probes, Inc., Eugene, OR) for 10 min at 38.5 °C. The spermatozoa pre-labeled with MitoTracker were resuspended in mTBM, and added to the 100 µL drops of mTBM for a final concentration of 2.5 to 5 × 10^4^ spermatozoa/mL. Matured oocytes were incubated with spermatozoa for 6 h at 38.5 °C, 5% CO_2_ in the air, then transferred to 500 µL drops of MU3 medium containing 0.3% BSA (A6003; Sigma) for additional culture.

### 2.5. Preparation of Porcine Oocyte Extracts

Cumulus cells from matured COCs were denuded with 0.1% hyaluronidase in TL-HEPES-PVA, and zonae pellucidae (ZP) were removed by protease (0.1%, *w*/*v*) (Pronase, Sigma) in TL-HEPES-PVA. The ZP-free, mature MII oocytes were transferred into an extraction buffer (50 mM KCl, 5 mM MgCl_2_, 5 mM ethylene glycol-bis(β-aminoethyl ether)-N,N,N′,N′-tetraacetic acid (EGTA), 2 mM β-mercaptoethanol, 0.1 mM PMSF, protease inhibitor cocktail (cat# 78410, ThermoFisher Scientific, Houston, TX, USA), 50 mM HEPES, pH = 7.6) containing an energy-regenerating system (2 mM ATP, 20 mM phosphocreatine, 20 U/mL creatine kinase, and 2 mM GTP), and submerged three times into liquid nitrogen for 5 min each. Next, the frozen-thawed oocytes were crushed by high-speed centrifugation at 13,000 rpm for 20 min at 4 °C in a Sorvall Biofuge Fresco (Kendro Laboratory Products). The supernatants were harvested, transferred into a 1.5 mL tube, and stored in a deep freezer (−80 °C). Unless otherwise noted, all chemicals used in this study were purchased from Sigma Chemical Co. (St. Louis, MO, USA).

### 2.6. Co-Incubation of Permeabilized Mammalian Spermatozoa with Porcine Oocyte Extracts

The permeabilized boar or bull spermatozoa were added to porcine oocyte extracts at a concentration of 1 × 10^4^ spermatozoa/10 μL of an extract derived from 2000 porcine oocytes/batch (200 μL) and co-incubated for 4–24 h in a humid atmosphere at 38.5 °C.

Mouse monoclonal anti-SQSTM1 antibody (cat. #ab56416; Abcam, Cambridge, MA, USA), previously shown to interfere with sperm mitophagy when injected in porcine zygotes [8], was added to the co-culture to examine the consequences of the inhibition of ooplasmic SQSTM1 binding to sperm mitochondria in the cell-free system. The permeabilized boar spermatozoa were co-incubated with porcine oocyte extracts at the same concentration as above for 4 h in a humid atmosphere at 38.5 °C, with or without the addition of 2 μg of anti-SQSTM1 antibody or control non-immune rabbit serum.

At the end of co-incubation, the spermatozoa exposed to oocyte extracts and the control spermatozoa were sedimented onto poly-L-lysine-coated 18 mm coverslips in a drop of KMT, pH = 7.0 at 38.5 °C on a slide warmer. The sperm-coated coverslips were fixed in 2% formaldehyde in PBS-NaN_3_ for 40 min at room temperature and used for immunocytochemistry immediately.

### 2.7. Immunocytochemistry to Visualize the Mitophagy Factors in The Cell-Free System

Spermatozoa affixed onto poly-L-lysine coated coverslips were permeabilized in PBS-NaN_3_ with 0.1% (*v*/*v*) Triton-X-100 (PBS-TX) for 40 min and blocked in PBS-TX containing 5% (*v*/*v*) normal goat serum for 25 min at the ambient temperature. The sperm-coated coverslips were immunolabeled with mouse monoclonal anti-SQSTM1 (cat. #ab56416; Abcam, Cambridge, MA, USA), rabbit polyclonal anti-VCP (cat. #sc-20799; Santa Cruz Biotechnology, Dallas, TX, USA), rabbit polyclonal anti-PACRG (Cat. #ab4090; Abcam), or rabbit monoclonal anti-SPATA18 (Cat. #ab180154; Abcam) antibodies overnight at 4 °C, followed by goat-anti-mouse (GAM)-IgG-FITC (1:200 dilution) or goat-anti-rabbit (GAR)-IgG-FITC (1:200) to demonstrate the localization of SQSTM1, VCP, PACRG, and SPATA18 in the cell-free system. Studies of PACRG were replicated with comparable results by using custom-produced monoclonal antibodies described previously [16]. Sperm DNA was counterstained with 2.5 μg/mL DAPI (Invitrogen-Molecular Probes, Eugene, OR, USA) added into the secondary antibody solution. Coverslips were mounted on microscopy slides in the VectaShield mounting medium (Vector Laboratories, Burlingame, CA, USA) and examined under a Nikon Eclipse 800 epifluorescence microscope (Nikon Instruments Inc., Melville, NY, USA) with a Retiga QI-R6 camera (Teledyne QImaging, Surrey, BC, Canada) operated by MetaMorph 7.10.2.240 software (Molecular Devices, San Jose, CA, USA).

### 2.8. Immunocytochemistry to Visualize the Mitophagy Factors in Oocytes and Zygotes

Oocytes and embryos were fixed in 2% formaldehyde for 40 min at room temperature. In some cases, the zona pellucida was removed using protease in TL-HEPES-PVA prior to fixation. Formaldehyde was then removed via washing in PBS. Oocytes/embryos were then permeabilized in PBS with 0.1% Triton-X-100 (PBS-TX) at room temperature for 40 min, then blocked in PBS-TX containing 5% normal goat serum (NGS) for 25 min. Oocytes/embryos were incubated with primary antibodies diluted in PBS containing 1% NGS and 0.1% Triton-X-100 overnight at 4 °C. The samples were incubated with a secondary antibody solution including 2.5 µg/mL DNA stain DAPI (Molecular Probes, Eugene, OR, USA), and goat-anti-rabbit (GAR)-IgG-FITC (1:100 dilution), GAR-IgG-TRITC (1:100), goat-anti-mouse (GAM)-IgG-FITC (1:100), or GAM-IgG-TRITC (1:100) for 40 min at room temperature. The oocytes were mounted on microscopy slides and photographed as described above for sperm samples.

### 2.9. SQSTM1 Fluorescence Intensity Measurement and Statistical Analysis

Images for pixel intensity measurements were acquired with identical magnification, exposure time, and acquisition settings for all treatment groups. The fluorescence intensity of SQSTM1 was analyzed by NIS-Elements software (v4, Nikon Inc., Melville, NY, USA). Values were represented as the mean of SQSTM1 fluorescence intensity ± SEM. Fluorescence intensities were evaluated by analysis of variance (ANOVA) using the SAS package (v9.3). Duncan’s multiple range test was used to compare significance between individual measurements when F-value was significant (*p* < 0.05).

### 2.10. SDS-PAGE and Western Blotting

Oocytes were suspended in 1× LDS loading buffer (106 mM Tris∙HCl, 141 mM Tris base, 2% (*w*/*v*) LDS, 10% (*v*/*v*) Glycerol, 0.75% (*w*/*v*) Coomasie Blue G250, 0.025% (*w*/*v*) Phenol Red, pH = 8.5) and sperm protein extracts were mixed with 4× LDS loading buffer in ratio 3:1, both supplemented with 2.5% β-mercaptoethanol. Spermatozoa were incubated at room temperature in 1× LDS loading buffer for 1 h on a rocking platform, spun, and the extract was supplemented with 2.5% β-mercaptoethanol. MII-oocytes and cultured embryos were incubated with protease to remove ZP and mixed with 1× LDS loading buffer supplemented with 2.5% β-mercaptoethanol. Prior to PAGE, all extracts were incubated at 70 °C for 10 min. Equal numbers of oocytes 50 or 100 per lane, as specified in images, spermatozoa (50 to 100 million per lane, as specified in images), and protein extract (the equivalent of 20 µg per lane) were loaded in each lane on a NuPAGE 4–12% Bis-Tris gel (Invitrogen, Waltham, MA, USA). Electrophoresis was carried out in Bis-Tris system using MOPS SDS running buffer (50 mM MOPS, 50 mM Tris base, 0.1% (*w*/*v*) SDS, 1 mM EDTA, pH = 7.7) with the cathode buffer supplemented with 5 mM sodium bisulfite. The molecular masses of separated proteins were estimated using Novex^®^ Sharp Pre-stained Protein Standard (cat # LC5800, Invitrogen, Carlsbad, CA, USA) run in parallel. PAGE was carried out for 5 min at 80 V to let the samples delve into the gel and then for another 60–70 min at 160 V. The power was limited to 20 W. After PAGE, proteins were electrotransferred to polyvinylidene fluoride (PVDF) membranes (Millipore) using Owl wet transfer system (Fischer Scientific, Waltham, MA, USA) at 300 mA for 90 min for immunodetection, using Bis-Tris-Bicine transfer buffer (25 mM Bis-Tris base, 25 mM Bicine, 1 mM EDTA, pH = 7.2) supplemented with 10% (*v*/*v*) methanol, and 2.5 mM sodium bisulfite. The membranes with the transferred proteins were blocked with 10% (*w*/*v*) non-fat milk in TBS with 0.05% (*v*/*v*) Tween 20 (TBST; Sigma-Aldrich) for 1 h and incubated with primary antibodies described in the Immunocytochemistry section, overnight at 4 °C and subsequently incubated with the HRP-conjugated goat anti-mouse IgG (GAM-IgG-HRP), or goat anti-rabbit (GAR-IgG-HRP) as secondary antibodies for 40 min at room temperature. The membranes were reacted with chemiluminescent substrate (Millipore Corporation, Billerica, MA, USA), detected using ChemiDoc Touch Imaging System (Bio-Rad, Hercules, CA, USA) to record the protein bands and analyzed by Image Lab Software (ver. 5.2.1, Bio-Rad, Hercules, CA, USA). The membranes were stained with CBB R-250 after chemiluminescence detection for protein load control.

## 3. Results

### 3.1. Sperm Priming for Cell-Free System Studies

To optimize the priming of sperm mitochondrial sheaths for oocyte extract co-incubation, boar spermatozoa pre-labeled with red fluorescent MitoTracker were treated with 0.05% lysolecithin for 10 min at 37 °C and subsequently incubated with different concentrations of DTT (0.0, 0.1, 1.0 or 10.0 mM) for 20 min at 37 °C. The spermatozoa in all treatment groups displayed uniform labeling of the mitochondrial sheaths pre-labeled with MitoTracker, and the strong, non-specific red fluorescence was detected in sperm heads due to non-specific MitoTracker uptake by sperm membranes and perinuclear theca (Figure 1A–E). Such labeling was not detrimental to experimental observation and resulted from the high initial MitoTracker concentration necessary for the specific mitochondrial sheath labeling to withstand the demembranation and permeabilization procedure. There were no obvious morphological changes in the sperm mitochondria after the incubation of spermatozoa with 0.05% lysolecithin alone (Figure 1B’,B’’), compared to control spermatozoa (Figure 1A’,A’’), despite the treatment causing the removal of the sperm plasma membrane. Incubation of spermatozoa with 0.05% lysolecithin followed by 0.1 mM DTT still showed intact sperm mitochondrial sheaths (Figure 1C’,C’’). Incubation of spermatozoa with 1 mM DTT for 20 min induced partial disintegration of mitochondrial sheaths, showing swollen and occasionally missing mitochondria/dented mitochondrial sheaths (Figure 1D’,D’’). A higher concentration of 10 mM DTT, combined with 0.05% lysolecithin, accelerated the stripping of the sperm midpiece/mitochondrial region (Figure 1E’,E’’). The latter two treatments also removed most of the MitoTracker labeling from the sperm tail midpiece. Altogether, the alterations of boar sperm mitochondrial sheath after demembranation and S-S reduction mirrored the observation of similarly primed bull and human spermatozoa [15]. As expected, a higher concentration of DTT also caused sperm flagella detachment.

### 3.2. Binding of Sperm Mitophagy Factors to Primed Sperm Tail Mitochondria in the Cell-Free System

To examine the recognition of sperm mitochondria by oocyte extract-derived mitophagy factors, boar spermatozoa were sequentially treated with 0.05% lysolecithin (10 min at 37 °C) and 10 mM DTT (20 min at 37 °C), and co-incubated with porcine oocyte extracts for 4 h, which regimen mimics early ooplasmic post-fertilization events. Spermatozoa exposed to oocyte extracts were immunolabeled with anti-SQSTM1 or anti-VCP antibodies. Ooplasmic SQSTM1, a known ubiquitin-binding autophagy receptor, was detected in the midpiece/mitochondrial sheaths of the demembranated sperm tails after co-incubation with oocyte extracts (Figure 2(Ab)), but not in the spermatozoa exposed for 4 h to extraction buffer (no oocyte extract) (Figure 2(Aa)). Such immunolabeling pattern indicated that SQSTM1 derived from oocyte extracts was able to recognize and bind to sperm mitochondria during co-incubation. This result is consistent with our previous observation of SQSTM1 associating with sperm mitochondria in the cytoplasm of porcine and rhesus monkey zygotes [8].

Protein dislocase VCP was detected in both the sperm mitochondrial sheath, the postacrosomal sheath, and the subacrosomal layer of the sperm heads (Figure 2(Ba)) in spermatozoa, which were not primed and were co-incubated with extraction buffer as control. After exposure to the sperm priming process, VCP was then detected in the sperm mitochondrial sheath only (Figure 2(Bb)); VCP was also prominent in the mitochondrial sheaths of spermatozoa which were primed and exposed to the oocyte extract for 4 h of co-incubation (Figure 2(Bc)). Such patterns of VCP localization in the cell-free system are consistent with previous observations of VCP associating with sperm mitochondria both before and after in vitro fertilization but dissipating from the sperm head after fertilization [8].

### 3.3. Investigation of Candidate, Sperm-Borne Mitophagy Determinants Using the Cell-Free System

After confirming SQSTM1’s ability to bind to sperm mitochondria in the cell-free system, we investigated two candidate mitophagy determinants, PACRG and SPATA18, known to integrate into sperm mitochondrial sheath and other sperm tail structures during spermiogenesis. Furthermore, both proteins have established roles in somatic cell mitophagy. Changes in localization pattern and/or labeling intensity of these proteins were assessed before and after the oocyte extract co-incubation using the cell-free system. Sperm mitophagy determinants such as these two would be expected to be recognized and degraded by the oocyte’s autophagic machinery early after fertilization. Thus, we anticipated observing changes in localization patterns of PACRG and SPATA18 after sperm-oocyte extract coincubation if they were, in fact, acting as mitophagic determinants.

PACRG was detected in ejaculated spermatozoa using WB at multiple molecular masses (Figure 3A) and found to localize to the mitochondria sheath (Figure 3B). Within our cell-free system control group, this mitochondrial sheath localization pattern was unaffected by sperm priming and 4 h of exposure to extract buffer, which lacked oocyte proteins (Figure 3C). However, spermatozoa that were exposed to oocyte proteins in the proper cell-free system for 4 h displayed a dramatic change in PACRG localization (Figure 3D). At this point, PACRG was found to be diminished in the mitochondrial sheath, but now detectable localize throughout the rest of the sperm tail and also in the sperm head post acrosomal sheath. After 24 h of cell-free system exposure, PACRG was still detectable within the sperm tail principal piece and throughout the head of the spermatozoa but absent from the sperm tail midpiece/mitochondrial sheath (Figure 3E). Finally, we screened for PACRG in a one-cell zygote, where it was no longer detectable on the mitochondrial sheath or the rest of the sperm tail (SQSTM1 was used as a mitochondrial sheath indicator) (Figure 3F).

SPATA18 was selected as a candidate pro-mitophagic protein because of its known function, implied by its sobriquet (mitochondria-eating protein), which is to degrade damaged mitochondria within a somatic cell. SPATA18 was detected in spermatozoa using Western blotting (Figure 4A) and immunocytochemistry (Figure 4B). SPATA18 was found throughout the sperm tail both before and after 4 h of oocyte extract exposure (Figure 4C,D). This localization shifted after 24 h of oocyte extract exposure and clearly became prominent on the mitochondrial sheath, while it was diminished, in a complementary fashion, in the sperm tail principal piece (Figure 4E). However, when observed in a one-cell zygote, neither localization pattern was detectable in the tails of the spermatozoa present in the fertilized oocyte cytoplasm (Figure 4F).

### 3.4. Sperm Head Expansion during Oocyte Extract Co-Incubation

To observe sperm morphological changes induced by prolonged exposure to oocyte extracts, boar sonicated and non-sonicated spermatozoa pretreated with 0.05% lysolecithin and 10 mM DTT (Figure 5A,A’,B,B’) were co-incubated with porcine oocyte extracts for 24 h (Figure 5C’,D’). The non-sonicated sperm heads exposed to oocyte extracts were enlarged after such prolonged incubation with extracts (Figure 5D’), whereas there were no morphological changes identifiable in the control sperm heads exposed only to oocyte extraction buffer (Figure 5D). Such sperm head expansion mirrored the early stages of paternal pronucleus formation observed in the zygote. Intriguingly, sperm heads isolated by sonication, and treated with lysolecithin-DTT, did not enlarge during prolonged culture in the absence of oocyte extract (Figure 5C,C’). It is possible that sonication disrupted the perinuclear theca and nuclear skeleton and also either removed or damaged resident factors required for the initial stages of paternal pronucleus development.

### 3.5. Inhibition of Ooplasmic SQSTM1 Binding to Sperm Mitochondria during Sperm-Oocyte Extract Co-Incubation

To determine if a conspecific antibody treatment prevents ooplasmic SQSTM1 binding to sperm mitochondria in the cell-free system, the permeabilized spermatozoa were co-incubated with oocyte extracts for 4 h, and an anti-SQSTM1 antibody was added to the mixture at the onset of co-incubation. This treatment prevented ooplasmic SQSTM1 from binding to sperm mitochondria (Figure 6(Aa)), while the SQSTM1 was detectable on sperm mitochondria after sperm-extract co-incubation without the addition of anti-SQSTM1 antibody (Figure 6(Ba)). Non-immune serum was used to generate negative controls for immunofluorescence and confirm treatment specificity (Figure 6(Ab,Bb)). Consistently, the intensity of SQSTM1-induced fluorescence labeling on the sperm midpiece/mitochondrial region was significantly lower (*p* < 0.05) after co-incubation with oocyte extract containing anti-SQSTM1 antibody (Figure 6C). This result agrees with the observations of porcine zygotes in which pre-injection with anti-SQSTM1 antibody blocked translocation of ooplasmic SQSTM1 to the sperm mitochondria post-fertilization [8].

### 3.6. Binding of Porcine Ooplasmic SQSTM1 to Bull Sperm Mitochondria in the Cell-Free System

We further examined if SQSTM1 derived from porcine oocyte extracts was detected in the interspecific cell-free system composed of porcine oocyte extract and bull spermatozoa. Non-treated (membrane-intact), control bull spermatozoa and bull spermatozoa treated with 0.05% lysolecithin and 10 mM DTT were immunostained with anti-SQSTM1 antibody. The SQSTM1 was not detectable in bull spermatozoa without exposure to porcine oocyte extracts (Figure 7A,A’,B,B’), whereas SQSTM1 originated from porcine oocyte extracts was visible in the midpiece/mitochondrial sheaths of bull sperm tails exposed to oocyte extract (Figure 7C,C’), suggesting that even in the interspecific conditions of the cell-free system, ooplasmic SQSTM1 was still able to recognize heterospecific sperm mitochondria.

## 4. Discussion

Using the cell-free system consisting of permeabilized, disulfide-reduced boar spermatozoa co-incubated with porcine oocyte extracts, we reconstituted the post-fertilization recognition of sperm mitochondria by autophagy/mitophagy factors SQSTM1 and VCP, as observed previously in mammalian (SQSTM1 and VCP [8]) and non-mammalian (SQSTM1 only) zygotes and embryos [17,18]. Cell-free extracts were derived from mature porcine MII oocytes. A volume of 10 μL extract derived from 2000 porcine oocytes/batch (200 μL) was sufficient to reconstitute the initial step of sperm mitophagy in 1 × 10^4^ boar spermatozoa primed by the demembranation and disulfide reduction, demonstrating that the oocyte extract can be used to study the mechanisms promoting clonal, maternal inheritance of mitochondria and mtDNA [8].

Boar sperm priming with lysolecithin and DTT, which mimics sperm plasma membrane removal during sperm oolemma fusion and subsequent disulfide reduction by ooplasmic factors, was sufficient to expose the sperm mitochondrial sheaths to the binding of autophagy factors from the oocyte extract. Sperm mitochondrial membrane proteins are hardened by disulfide cross-linking [15,19,20,21]. By using disulfide bond-reducing agents such as DTT and β-mercaptoethanol, sperm mitochondria can be completely dissociated from the sperm tail structure [22]. The stabilization of sperm accessory structures by disulfide bridges may be a unique feature of mammalian spermatozoa [23]. Invertebrates and lower vertebrates, including sea urchin and *Xenopus laevis* spermatozoa, lack the disulfide stabilization of sperm accessory structures, showing no structural changes after treatment with disulfide bond reducing agents [15,24]. Experiments aimed at optimizing sperm priming prior to the extract co-incubation indicate that disulfide reduction is necessary for establishing a porcine oocyte cell-free system.

Prominent among mammalian sperm structures stabilized with disulfide bridges is the sperm nucleus in which protamines become cross-linked during epididymal sperm maturation [15,25]. Protamines are arginine-rich DNA binding proteins that supplant nuclear histones during spermatid differentiation to mediate the hypercondensation and packaging of sperm DNA into a compact sperm nucleus [26]. Sperm nucleus protamination is reversed at fertilization during the disassembly of sperm accessory structures at the onset of embryonic development. During natural fertilization, the reduction in disulfide bonds in the cysteine residues of protamines facilitates sperm nuclear decondensation, male pronuclear development, and replacement of protamines with oocyte-derived histones [27]. This protamination reversal is mediated by oocyte-produced glutathione (the tripeptide γ-glutamyl-cysteinyl-glycine; GSH) [28] and may facilitate proteasomal degradation of sperm nuclear protamines, as proteasomal inhibitors hinder paternal pronuclear development [29]. The above findings support our observation in which sperm head expansion was induced by an oocyte extract following sperm priming by demembranation and disulfide bond reduction, thus mimicking the early stages of paternal pronucleus formation in the zygote. This sperm priming process certainly may influence the localization of some spermatozoa proteins. However, this shift in localization should be mimicking physiological events because both plasma membrane removal and disulfide bond reduction are notable in vivo fertilization events.

Ubiquitin-binding autophagy receptor SQSTM1 links ubiquitinated proteins to the autophagic pathway [30]. The SQSTM1 binds to polyubiquitinated proteins through its C-terminal polyubiquitin-binding UBA domain [31]. Mammalian sperm mitochondria carry (poly)ubiquitinated proteins of spermatogenic origin, while additional mitochondrial substrates may become ubiquitinated after sperm incorporation in the ooplasm at fertilization [6,7,32]. Specifically, sperm mitochondrial membrane protein prohibitin (PHB) and mitochondrial transcription factor A (TFAM) are ubiquitinated in bull and boar spermatozoa, respectively [32,33]. Our recent study identified three SQSTM1/UBA domain co-purifying boar sperm mitochondrial proteins, including mitochondrial trifunctional enzyme subunit alpha/HADHA, mitochondrial aconitase ACO2, and mitochondrial ATP synthase H^+^ transporting F1 complex β-subunit/ATPF1B [8]. These and other mitochondrial proteins are plausible binding partners/substrates of SQSTM1 during post-fertilization sperm mitophagy. Our previous finding that SQSTM1 derived from the oocyte exclusively associated with sperm mitochondria in porcine and rhesus monkey zygotes [8] was replicated in a cell-free system in the present study. In addition, the microinjection of autophagy-targeting antibodies specific to SQSTM1 and GABARAP interfered with mammalian sperm mitophagy [8], and the addition of anti-SQSMT1 antibody to sperm-oocyte extract mixtures consistently prevented the oocyte-cytosolic SQSTM1 from binding to sperm mitochondria in the present study. Parallel to sperm mitochondrion changes, the sperm heads in the present study became enlarged during the prolonged incubation in oocyte extract, mimicking the initial steps of paternal pronucleus development. The addition of anti-SQSTM1 antibody to the sperm-oocyte extract mixture prevented the binding of ooplasmic SQSTM1 to sperm mitochondria, again mimicking the events of early zygotic development. In addition, SQSTM1 derived from porcine oocyte extracts was detected in bull sperm mitochondria, showing interspecific recognition of ooplasmic SQSTM1.

The VCP protein belongs to the AAA+ (ATPase associated with diverse cellular activities) protein family that participates in various cellular events, including cell division, protein degradation by UPS, mitochondria-associated protein degradation, and autophagosome maturation [34,35]. Acting as a protein dislocase, VCP extracts ubiquitinated mitochondrial membrane proteins and presents them to the 26S proteasome for degradation during organelle renewal mitophagy in somatic cells [35]. Association of the VCP with sperm mitochondria exposed to oocyte extracts agrees with previous studies showing that VCP protein was detectable in the mitochondrial sheaths of boar spermatozoa and remained associated with sperm mitochondria after fertilization [8].

In addition to SQSTM1 and VCP, we used our complementary cell-free and IVF systems to explore the post-fertilization fate of pro-mitophagic proteins, PACRG and SPATA18. Parkin co-regulated gene product, PACRG, is regulated by the UPS. The E3 type ubiquitin ligases which regulate it have yet to be identified [36], though the components of the linear ubiquitin chain assembly complex (LUBAC), including ubiquitin ligases HOIP and HOIL-1L, are plausible candidates [37]. The formation of ubiquitinated protein aggregates, the autophagy-targeted aggresome particles, is mediated by PACRG, particularly under conditions of repressed proteasomal proteolysis. These PACRG-containing aggresomes are then removed through macroautophagic pathways. In this sense, mitochondrial sheaths of mammalian spermatozoa are similar to aggresomes and are detectable using aggresome-detecting probes [8,38]. Thus, the whole sperm mitochondrial sheath may be recognized as a large aggresome by the oocyte autophagy machinery. PACRG was selected as a target protein in the present study because it shares a bidirectional promoter with the gene *parkin*, which codes for the PRKN protein [8]. The PRKN protein plays a role in a well-defined somatic cell mitophagic pathway, the PINK1-PRKN mitophagic axis [39], which contributes to post-fertilization sperm mitophagy in the mouse zygotes, in synergy with the SQSTM1-regulated mitophagy branch [40]. The identification of PACRG on the mitochondrial sheath prior to fertilization and throughout the tail after 4 and 24 h of oocyte extract exposure could build upon this theory. PACRG may serve as a sperm-borne substrate that signals for aggresome-clearing autophagy of the mitochondrial sheath. Further supporting this theory, it has been shown that PACRG levels are directly correlated with the activity of certain autophagic pathway branches. In particular, PACRG and SQSTM1 (alias P62/p62) directly interact following proteasomal inhibition. However, an increase in PACRG alone did not result in aggresome formation; the inhibition of the proteasome was necessary, indicating that PACRG seems to function in a compensatory role [41]. Perhaps this PACRG pathway is also used during early fertilization to degrade the sperm tail and the mitochondrial sheath through autophagy. PACRG has also been shown to be crucial for spermiogenesis, and its carryover in the sperm mitochondria may reflect such function. Male PACRG knock-out mice are unable to assemble sperm flagellum and undergo spermiogenesis arrest [42]. PACRG mediates both the formation of the sperm tail [42] and somatic cell aggresomes [43] via interactions with the microtubule network. Mutations in the *PACRG* gene have been implicated in the etiology of human asthenozoospermia [44]. The progressive change in localization of sperm PACRG after the oocyte extract treatment in this study (Figure 3), which resulted in its disappearance at the one-cell stage after fertilization, may help elucidate its function within the context of post-fertilization sperm mitophagy. The localization of PACRG to the mitochondrial sheath of freshly ejaculated spermatozoa appears to be a remanent of its role in sperm tail formation (Figure 3B), with perhaps a dual function related to post-fertilization sperm mitophagy. It was observed after 4 and 24 h of the cell-free system exposure that the presumed oocyte-derived PACRG molecules appeared to localize throughout the sperm tail principal piece tail and the post-acrosomal region of the sperm head (Figure 3D,E). In somatic cells, PACRG plays a role in aggresome formation around mitochondria and then plays a role in signaling for degradation. The sperm tail localization pattern of PACRG after 4 and 24 h of cell-free system exposure may be evidence of PACRG acting in conjunction with UPS and signal to recruit SQSTM1, which could explain the absence of PACRG from sperm mitochondria in a one-cell embryo where SQSTM1 has clearly relocated to the sperm mitochondrial sheath (Figure 3F). Comparatively lesser intensity of SQSTM1 and incomplete removal of PACRG is observed after sperm exposure to oocyte extracts in the cell-free system (present study), which may also lack some of the organelle membranes and lysosomes necessary for the completion of sperm mitophagy. Our future experiments will thus include varied proportions of oocyte cytosol and membrane fractions used in the cell-free system.

Mitochondria eating protein SPATA18 serves for mitochondrial quality control in various cell types. It mediates the degradation of mitochondria using vacuole-like structures, which engulf and degrade unhealthy mitochondria [45]. The SPATA18 protein can facilitate mitophagy because of the recognition of DNA damage through the p53-SPATA18 axis. This axis serves as a mitochondrial homeostasis maintaining mechanism rather than a direct DNA damage response. SPATA18 induces the formation of intramitochondrial lysosome-like organelle after mitochondrial damage, which seem to act independently of conventional autophagy [46]. An increase in mitochondrial ATP output under stress leads to the mitochondria producing more reactive oxygen species (ROS), which, in turn, increases mitochondrial stress. This is where SPATA18 comes into play as a mitochondrial quality control measure. Likewise, sperm mitochondria face stress and possible mtDNA damage by ROS as the spermatozoa journey across the male and female reproductive tracts to reach the site of fertilization in the oviduct [45]. Perhaps SPATA18 works along the mitochondrial quality control axis on the sperm mitochondria upon entry into the oocyte cytoplasm, assuring that potentially damaged paternal mtDNA is not passed onto the next generation. Paternal mtDNA transmission to the offspring has been observed in various interspecific hybrids throughout early embryonic development, whereas sperm mtDNA was eliminated in intraspecific hybrids [6,47,48,49]. Therefore, mitophagy machinery may be insufficient to eliminate sperm mitochondria in interspecific crosses, although our observation showed bull sperm mitochondria were recognized by the mitophagy factor SQSTM1 in the interspecific cell-free system. Such a system could be used in the near future to identify factors responsible for the species specificity of mammalian sperm mitophagy.

## 5. Conclusions

The present study provides the first evidence that mammalian sperm mitophagy can be reconstituted in a relevant conspecific cell-free system. Sperm mitophagy factors SQSTM1 and VCP were detected in sperm mitochondria after oocyte extract co-incubation, recapitulating the early events of sperm mitophagy observed in mammalian zygotes. Candidate sperm mitophagy factors PACRG and SPATA18 were detected in spermatozoa before and after cell-free system exposure, and the changes in their localization were further investigated in one-cell embryos. This set of experiments highlights the usefulness of the cell-free system in the investigation of the sperm-borne mitophagy factors, especially those mitophagy determinants lost early after fertilization and by the time of pronuclear formation. Altogether, our observations demonstrate that the developmental events of post-fertilization sperm mitophagy observed in porcine zygotes can be reconstituted in a cell-free system, which could become a useful tool for identifying additional molecules that regulate the mitochondrial inheritance in mammals. Experiments are underway, using quantitative proteomics to compare the sperm proteomes before and after oocyte extract exposure, with particular focus on sperm mitochondrial proteins lost during coincubation and oocyte proteins associating with the sperm mitochondrial sheaths at the same time.

## Figures and Tables

**Figure 1 cells-10-02450-f001:**
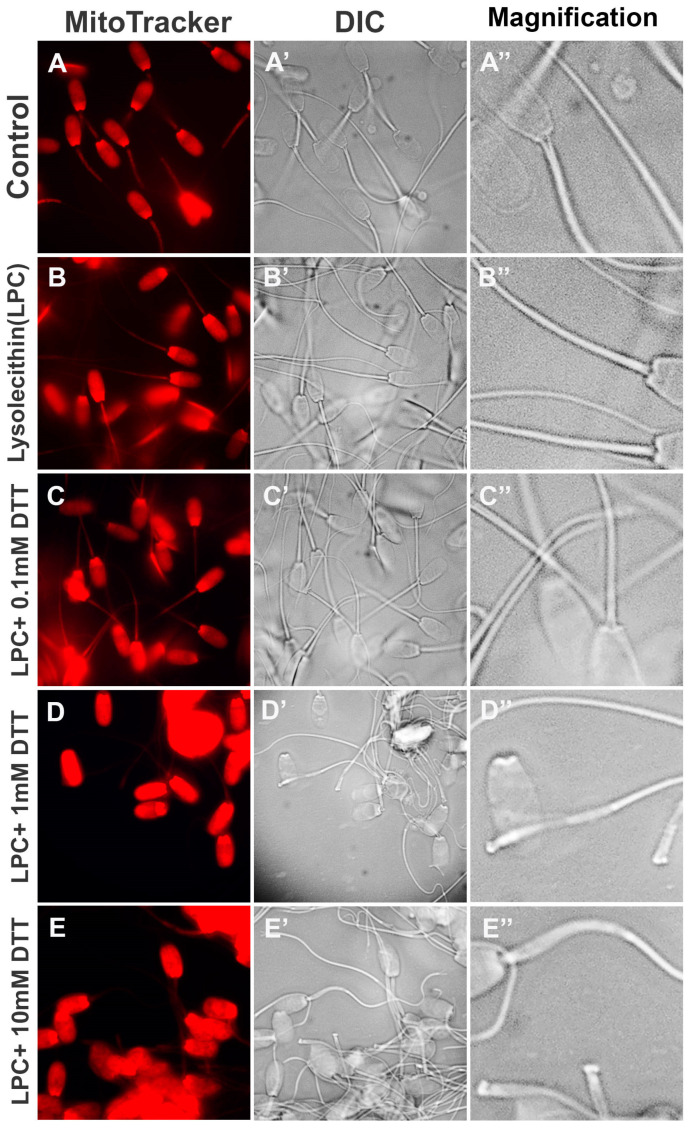
Sequential demembranation and reduction in sperm protein disulfide bonds prime sperm mitochondria for co-incubation with oocyte extracts. Spermatozoa, pre-labeled with MitoTracker, were demembranated with 0.05% lysolecithin and subsequently incubated with various DTT concentrations (0.0, 0.1, 1.0, and 10.0 mM) (**A**–**E**). The morphological changes of sperm mitochondria were not obvious after the incubation of spermatozoa with 0.05% lysolecithin alone (**B’**,**B’’**) compared to membrane-intact control spermatozoa (**A’**,**A’’**). Incubation with 0.05% lysolecithin followed by 0.1 mM DTT still showed intact sperm mitochondrial sheaths (**C’**,**C’’**), while incubation with 1 mM DTT for 20 min induced partial disintegration of mitochondrial sheaths (**D’**,**D’’**). A high concentration of 10 mM DTT accelerated the stripping of mitochondria from sperm tail midpiece/mitochondrial sheaths (**E’**,**E’’**). Original magnification: 1000×.

**Figure 2 cells-10-02450-f002:**
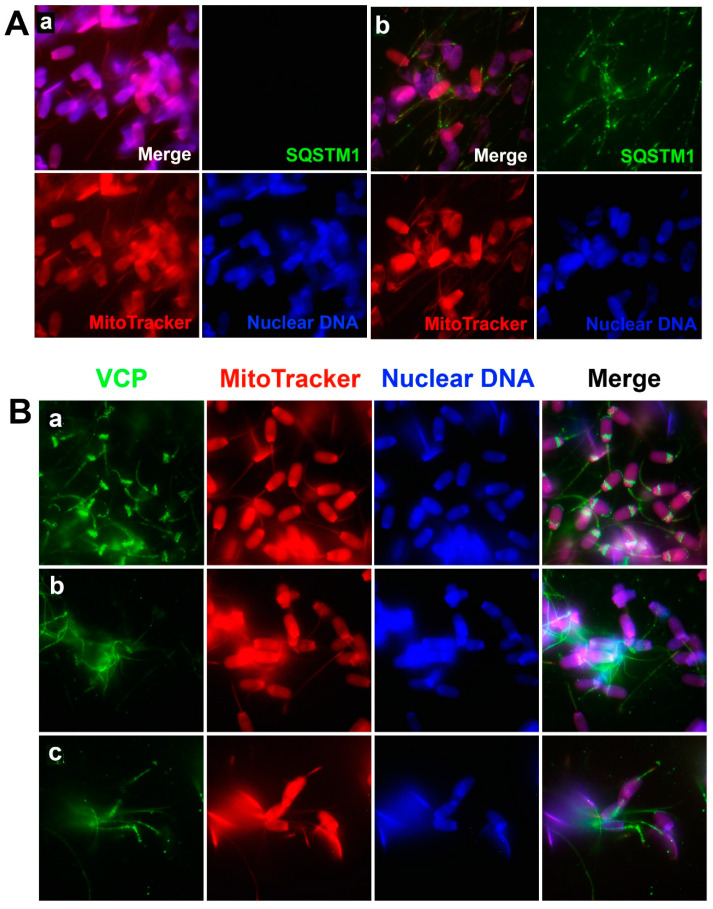
Binding of mitophagy factors SQSTM1 and VCP to sperm mitochondria in the porcine cell-free system. Spermatozoa pre-labeled with MitoTracker were primed with 0.05% lysolecithin and 10 mM DTT, then co-incubated for 4 h with porcine oocyte extracts or with an oocyte extraction buffer (no oocyte extract) as a control, fixed and immunolabeled with anti-SQSTM1 (**A**) or anti-VCP antibodies (**B**). SQSTM1 was not detected in spermatozoa exposed solely to the extraction buffer (**Aa**), but SQSTM1 was detected in the midpiece/mitochondrial sheaths of sperm after co-incubation in oocyte extract (**Ab**). VCP was detected in spermatozoa that were not exposed to the lysolecithin or DTT treatment and were exposed to extraction buffer (no oocyte extract); these spermatozoa acted as a control (**Ba**). Likewise, VCP was detected in spermatozoa that were then exposed to the lysolecithin and DTT priming process and co-incubated for 4 h with extraction buffer (no oocyte extract) (**Bb**) and also oocyte extract (**Bc**). The VCP protein was prominent in the sperm mitochondrial sheath both before (**Bb**) and after (**Bc**) co-incubation, while the control spermatozoa without treatment of DTT and oocyte extracts also displayed VCP in postacrosomal sheaths and the subacrosomal layer of the sperm head (**Ba**). Original magnification: 1000×.

**Figure 3 cells-10-02450-f003:**
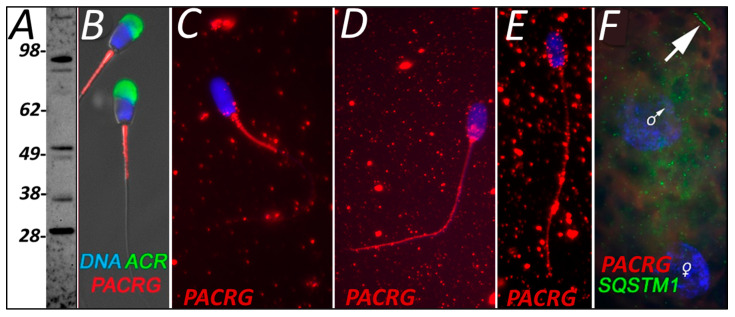
PACRG was identified in ejaculated sperm utilizing Western blot detection (**A**) and immunocytochemistry before fertilization (**B**). Predicted mass for various PACRG isoforms ranges from 26.5 to 33 kDa, while polymorphisms and stable, covalent complexes with chaperones and other proteins are known, including covalent modification by multi-ubiquitin chains. Within the cell-free system, primed and buffer exposed spermatozoa saw no difference in localization when compared to ejaculated sperm (**C**); however, after 4 h of cell-free treatment PACRG underwent a dramatic change in localization (**D**). After 24 h of cell-free system exposure, a similar localization signature was observed (**E**). PACRG was no longer detectable in a one-cell zygote (**F**).

**Figure 4 cells-10-02450-f004:**
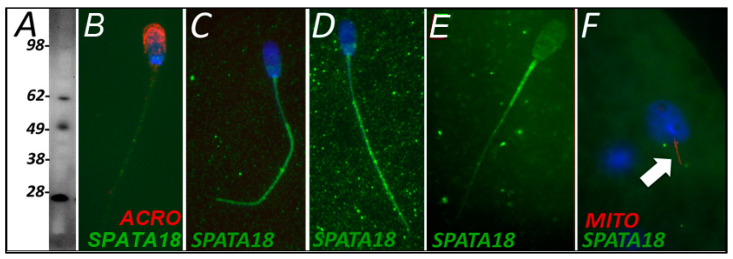
SPATA18 was detected in freshly ejaculated, and cell-free system treated spermatozoa (**A**–**E**) and zygotes (**F**) using Western blotting (**A**) and immunocytochemistry (**B**–**E**). (**A**) Western blotting detected a band consistent with the predicted mass in *S. scrofa* (59 kDa), as well as several lower bands likely resulting from protein degradation. Fresh ejaculated spermatozoa appeared to have the tail and postacrosomal sheath labeling present (**B**). Within the cell-free system, primed and buffer exposed spermatozoa appeared to have SPATA18 localized throughout the tail but mainly to its principal piece (**C**). After 4 h of cell-free exposure, spermatozoa displayed a seemingly identical localization pattern (**D**). After 24 h of cell-free system exposure, the localization pattern shifted from sperm tail principal piece to midpiece/mitochondrial sheath (**E**). However, SPATA18 was no longer detectable in a one-cell zygote (**F**).

**Figure 5 cells-10-02450-f005:**
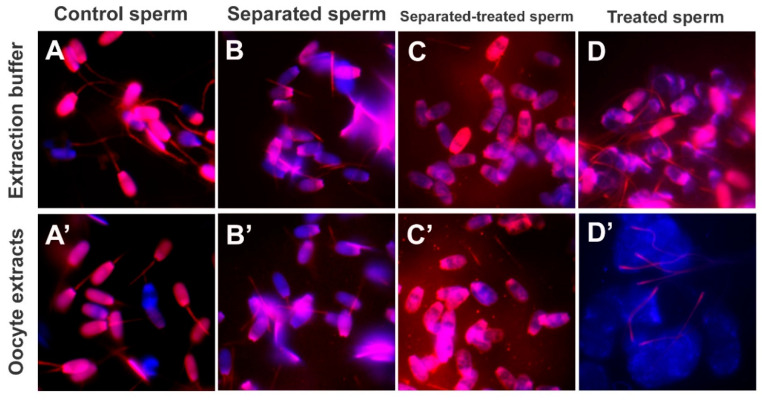
Sperm head expansion during oocyte extract co-incubation. Intact spermatozoa (**A**,**A’**) and sperm heads and tails separated by sonication (**B**) were incubated with oocyte extracts or with extraction buffer as a control. Both the intact spermatozoa and the separated sperm heads and tails were subsequently treated with lysolecithin and DTT and co-incubated with porcine oocyte extracts for 24 h (**C**,**D**). The control sperm fractions incubated with oocyte extraction buffer (**D**) did not show signs of sperm head expansion, but the heads of spermatozoa exposed to oocyte extracts appeared enlarged (**D’**) at the end of incubation. Sperm heads separated by sonication did not swell during prolonged incubation with oocyte extracts (**B’**,**C’**). Original magnification: 1000×.

**Figure 6 cells-10-02450-f006:**
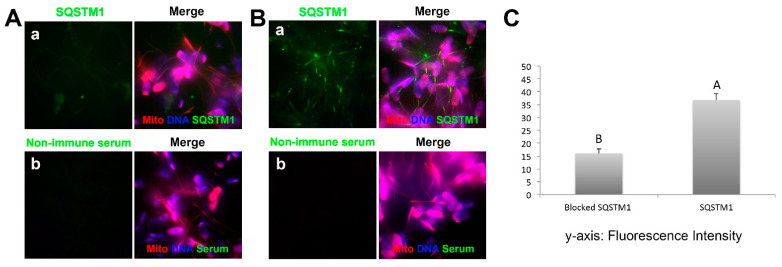
Inclusion of an isospecific antibody into sperm-oocyte extract co-incubation mixture prevents the binding of ooplasmic SQSTM1 to sperm mitochondria. Spermatozoa treated with lysolecithin and DTT were co-incubated with porcine oocyte extracts for 4 h with or without the addition of anti-SQSTM1 antibody, then washed and processed for immunofluorescence with the same antibody or with non-immune mouse serum (control). Such treatment prevented oocyte-extract derived SQSTM1 from binding to sperm mitochondria (**Aa**), while in a positive control sample (control mouse serum addition to the extract), SQSTM1 was detectable on sperm mitochondria at the end of sperm-extract co-incubation (**Ba**). No significant signals were detected with non-immune serum in negative control for immunofluorescence (**Ab**,**Bb**). Original magnification: 1000×. (**C**) Relative pixel intensity values of SQSTM1 fluorescence in the sperm mitochondrial sheath after sperm-oocyte extract co-incubation with/without the addition of anti-SQSTM1 antibody. The fluorescence intensity of ooplasmic SQSTM1 binding to sperm mitochondria was significantly lower in sperm-extract co-incubation blocked with an anti-SQSTM1 antibody. Values are expressed as the mean of fluorescence intensity ± SEM. Different superscripts A and B in the diagram denote significant differences at *p* < 0.05.

**Figure 7 cells-10-02450-f007:**
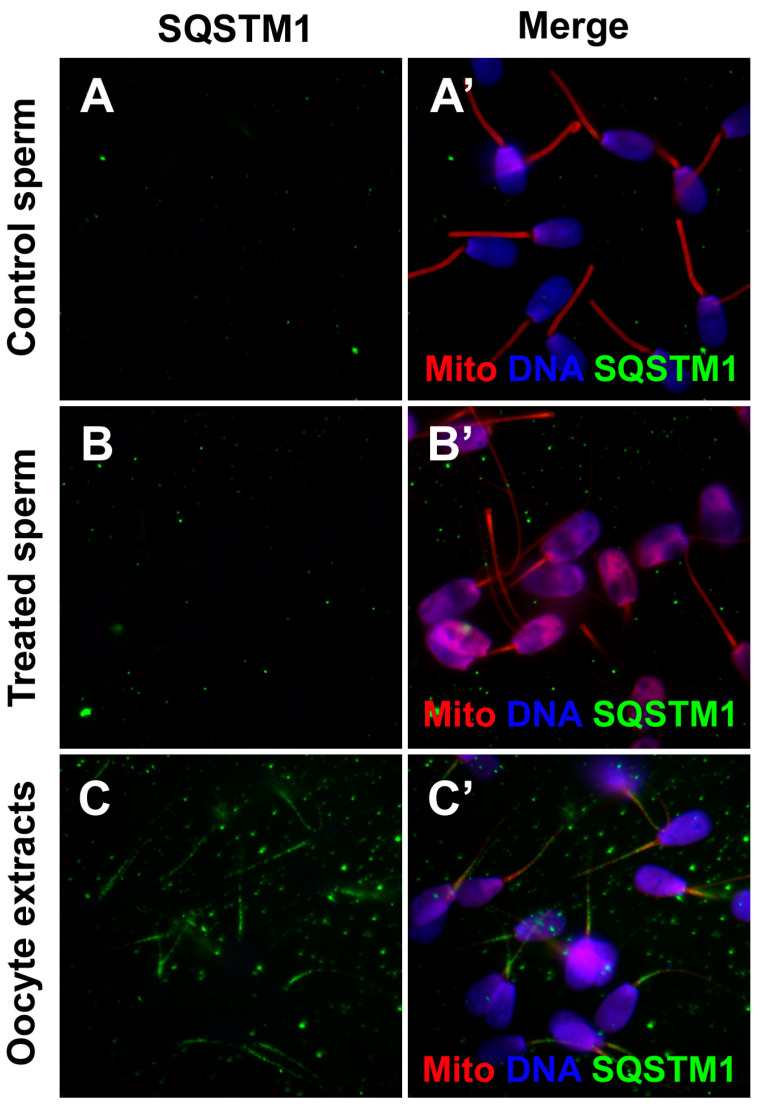
Binding of ooplasmic SQSTM1 to bull sperm mitochondria co-incubated with porcine oocyte extracts. Control bull spermatozoa pre-labeled with MitoTracker (**A**,**A’**) and bull spermatozoa primed with 0.05% lysolecithin and 10 mM DTT (**B**,**B’**) were immunolabeled with anti-SQSTM1 antibody. There was no SQSTM1 labeling in bull sperm mitochondria prior to co-incubation. (**C**,**C’**), Spermatozoa primed with 0.05% lysolecithin and 10 mM DTT and co-incubated with porcine oocyte extracts for 4 h, showed immunolabeling with anti-SQSTM1 antibody. Original magnification: 1000×.

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
