# Peer review of "Mammalian Cell-Free System Recapitulates the Early Events of Post-Fertilization Sperm Mitophagy"

_cells, 2021, doi:10.3390/cells10092450_

Round 1
Reviewer 1 Report
The authors performed a new method to study sperm mitophagy after fertilization. Thus, the authors could demonstrate that some proteins previously studied act similarly in a cell-free system. Thus, the results added new method that would help to observe the effects in sperm mitophagy and fertilization in a sperm-focused study.
I just would like to add a few comments to the study:
- Introduction: A longer explanation about cell-free system should be added to the new audience.
- Methods: line 112 - the sperm were kept at the storage until 5 days at room temperature? is this information correct?
- Methods: Sperm priming: DTT concentration were increasing each step or different concentrations of DTT were tested? I think this information was not so clear in this section.
- Figures: The images are very suitable, but sometimes there are som many group that it makes confusing to follow. I would recommend to always let in the first line, the "clean" controls, with no treatment, and then the following groups, adding some grade of difficulty. Specifically, figure 6 should be grouped differently.
Reviewer 2 Report
This is a piece of work that clearly derivates from the earlier published by the group in Proc Natl Acad Sci (2016 Sep 6;113(36): E5261-70). In the present manuscript the authors established a cell-free system to study the events related to post-fertilization sperm mitophagy. This is a good work, but I have some concerns about the methodology that supports the conclusions.
Main comments
- In line 164 the extraction buffer was established as follows: extraction buffer [50 mM KCl, 5 mM MgCl2, 5 mM ethylene glycol-bis(β-aminoethyl ether)-N,N,N’,N’-tetraacetic acid (EGTA), 2 mM β-mercaptoethanol, 0.1 mM PMSF, protease inhibitor cocktail (cat# 78410, ThermoFisher Scientific, Houston, TX), 50 mM HEPES, pH = 7.6].
Can EGTA, PMSF, and protease inhibitor cocktail, present in the extraction buffer, interfere with the correct functions of oocyte proteins in the further applications?
According to the hypothesis that oocyte proteins should acts on sperm structures , those buffer compounds could interfere with the proper function of oocyte proteins. At least, oocyte proteins with protease functions and proteins that needs calcium for their correct function would be affected. Therefore, proteins that help to decondense the male nucleus and mitochondrial sheath will not be able to execute their functions when co-incubated with sperm?
Although it may be a harder work, I think that it would be more proper to obtain oocyte extracts by physical methods such as strokes on Glass/Teflon Potter Elvehjem homogenizers and ultracentrifugation, and to use these extracts immediately. Sperm instead are stored on extender and can be used as desired.
- Were sperm directly resuspended in 10 ml of oocyte extract after priming with 0.05% (w/v) lysolecithin and DTT? Or were the sperm suspended in a specific volume of KMT supplemented with 0.05% (w/v) lysolecithin and DTT and then joined? If the last is correct, which was the volume of the suspension?
The remanent DTT can also affect the correct function of oocyte proteins involved in post-fertilization events.
The relocation of VCP protein (to the subacrosomal region) showed after incubation with the oocyte extraction buffer may be indicative that the buffer affects the protein functions. Line 286-324. Figure 2Bc.
It can be speculated that the changes in localization of SPATA18 and PACRG after coincubation could be due to buffer components as well as to oocyte derived compounds.
- The specificity of the antibodies is questionable. It is stated that anti-PACRG antibodies recognize a protein of predicted mass ranging from 26.5 to 33 kDa. But in Fig. 3A several bands of higher molecular mass appear in the Western blot. The same situation occurs for SPATA18. Moreover, this antibody recognizes a band of lower mass than the expected for the studied protein with higher intensity. RW Burry, J Histochem Cytochem. 2011 Jan; 59(1): 6-12), established a guide for specificity of antibodies that may be consulted.
- Line 322. Protein dislocase VCP was detected in sperm mitochondrial sheaths after extract co-incubation (Figure 2 B; a); VCP was also prominent in spermatozoa permeabilized with lysolecithin and DTT, but not exposed to oocyte extracts
However, there is a clear signal in 2Bb (oocyte extraction buffer (no oocyte extract) as control (B; b))
Minor comments
- I think immunocytochemistry is not appropriated term. There is no chemical reaction to demonstrate the presence of the protein. I think immunofluorescence would be more proper.
- In order to demonstrate mitochondrial sheath labelling to withstand the demembranation and permeabilization procedure, the probe MitoTracker® Red CMXRos was used. Once demonstrated, I think that the use of this probe in subsequent experiments is unnecessary and adds background to images
- Line 264. There were no obvious morphological changes in the sperm mitochondria after the incubation of spermatozoa.
This affirmation seems incorrect. To demonstrate morphological changes in the sperm mitochondria, electron microscopy must be used.
The same for line 292
- Fig 3B and 4B. ACR-labelled sperm is shown with two different colors in the respective figures. Which probe was used to label acrosomes?
- Line 440: 10 μl extract derived from 2,000 porcine oocytes/batch
The sentence starts with a number instead of the word “Ten”
- Line 536: It was observed after 4 and 24 hours of the cell-free system exposure that the presumed oocyte-derived PACRG molecules appeared to localize throughout the sperm tail principal piece tail and the post-acrosomal region of the sperm head
I think PACRG is not a oocyte-derived molecule
Author Response
Please see the attachemnet.

Round 2
Reviewer 2 Report
All my concerns have been addressed.